# Smart Villages Policies: Past, Present and Future

Simona Stojanova [1], Gianluca Lentini [2], Peter Niederer [3], Thomas Egger [3], Nina Cvar [1], Andrej Kos [1] and Emilija Stojmenova Duh [1,*]

[1] Faculty of Electrical Engineering, University of Ljubljana, Tržaška Cesta 25, 1000 Ljubljana, Slovenia; simona.stojanova@fe.uni-lj.si (S.S.); nina.cvar@fe.uni-lj.si (N.C.); andrej.kos@fe.uni-lj.si (A.K.)
[2] Poliedra-Politecnico di Milano, via G. Colombo, 40, 20133 Milan, Italy; gianluca.lentini@polimi.it
[3] Swiss Center for Mountain Regions (SAB), Seilerstrasse 4, 3001 Berne, Switzerland; peter.niederer@sab.ch (P.N.); thomas.egger@sab.ch (T.E.)
* Correspondence: emilija.stojmenova@fe.uni-lj.si

**Abstract:** Highlighting the important role of rural development, this paper represents a review of rural policies. Data were generated, including a broad literature research and online survey on existing and future, post-2020 rural development policies. The survey was shared among project partners from six different EU Member States including eleven regions, all from the Alpine Space. The number of total policies covered in this review paper, together with policy projects, programs or actions, is 114. Based on these, key policy findings and future recommendations are provided, attributing to the future studies on this topic as well as for policymakers at the local, regional, national and EU levels.

**Keywords:** sustainability; rural development; rural policies; Smart Villages; smart development; digital transition

## 1. Introduction

In addition to the general global trend of migration to urban places, a great number of people around the world still live in peripheral areas. According to EUROSTAT and data from 2020, this number in the European Union (EU) is 29% of the whole population living in the EU [1]. At the same time, these areas fall behind in providing the services of general interest such as health, education, and transportation, or are not at the same level as those provided in the urban areas [2]. The rural–urban gap today is more visible than ever and village life is still not that attractive for people, especially for the younger generations. Research on Smart Villages aims to discover the issues and problems in the everyday life of rural inhabitants, to address these challenges and find a solution for them by using the advantages from digitalization and Information and Communication Technologies (ICTs) [3].

In order to reduce these inequalities and improve rural residents' lives, the use of technology and access to internet connection is essential nowadays, rather than a choice of preference [4]. ICT is an umbrella term that covers different ranges of technology, from the very simple, such as phone text messaging services, to very advanced, such as advanced software solutions. Because of its affordability, ICTs have been applied also within rural areas and the agricultural sector, which is the most commonly spread sector in rural areas. The advantages of using ICTs in agriculture can be seen in terms of providing an inexpensive means of connectivity and affordable implementation, online data storage and transfer, the emergence of new efficient business models, new ways of cooperation and an increased demand for agricultural information services [5].

As part of this global digitalization, the Internet of Things (IoT)—or the process of connecting physical things to the internet and creating remote control—facilitates the process of data transfer and secure information sharing. Furthermore, using IoT can lead to the general improvement of rural living standards [6].

The benefits of investing in Smart Villages' development can be said to be two-fold. They benefit the inhabitants, making their life more comfortable and straightforward, but they also make a community more empowered, resilient, independent and connected. Furthermore, Smart Villages are also contributing to the more efficient use of available resources. This also develops the idea of community and belonging, which in turn can make these regions more autonomous and independent [7].

Although the concept of a Smart Village is most frequently linked to agriculture, it goes far beyond it. It represents a broader concept, covering heterogeneous activities, mainly supported and empowered technology, that act as tools for improving services in rural areas and make the rural functioning easier and smoother, leading to the overall enhancement of rural village dwellers [8].

In defining Smart Villages, the European Commission (EC) refers to rural communities which use their current advantages and strive for their digital development whilst simultaneously supporting everyday activities related to an increased quality of life and living standard, rethinking general public services and considering environmental aspects in every action [9]. In the same manner, ENRD defines this concept as: "Smart Villages are communities in rural areas that use innovative solutions to improve their resilience, building on local strengths and opportunities. They rely on a participatory approach to develop and implement their strategy to improve their economic, social and/or environmental conditions, in particular by mobilizing solutions offered by digital technologies" [10].

The EU Action for Smart Villages was introduced in 2017 by the EU as a policy concept, where the focus is on the implementation of rural development strategies and actions as an integrated community level approach of collectivism [11]. Combining the data from existing literature studies and the data generated from the online survey, we can define the research goals of this paper:

- Based on the postulates from the Sustainable Hub to Engage into Rural Policies with Actors (SHERPA) project (2020) [12] and the OECD framework for rural development [13], this paper determines and explores six areas, which are identified as the most important for rural development. This research aims for their review and analysis.
- Deriving from the work of Visvizi, Lytras and Mudri [14], where they present an integrated approach to sustainable EU Smart Villages policies and Zavratnik, Kos and Stojmenova Duh [15] and their reviews of current Smart Village (SV) initiatives and practices, this research will systematize current policies/projects/actions with a focus on the future, related, most known and least known policies/projects/actions.
- Comparison between Smart Villages and rural development policies in six member states involved in the Interreg Alpine Space Smart Villages project, followed by further policy recommendations at the local, national and EU levels.
- In short, we can formulate the following research objects:
- Analyze the most important areas for rural development.
- Smart Villages and rural development policies characteristics analysis.
- Provide recommendations for the improvement of Smart Villages and rural development policies.

The paper is structured as follows. Firstly, the state of the art of Smart Villages implementation and funding organizations is presented. Subsequently, in the research design and methodology, we describe the rural research areas and categorization, followed by data presentation from the literature and data collected from our questionnaire. Then, we present the data analysis and discussions, highlighting the main study findings and we finish with the concluding remarks and the availability of materials or information.

## 2. State of the Art

The postulation of 'sustainable development' originates from a discipline in economics by an English economist Thomas Malthus who started reasoning the lasting of natural resources [16]. If Thomas Malthus was the first to talk about sustainability from the economical point of view, German explorer Alexander von Humboldt is often credited to be the first scientist to overcome the traditional view of environment as something subservient to mankind, and to theorize the need for humanity to respect nature's integrity and complex balance [17]. His concept of 'Naturgemälde', or 'Portrait of nature' is the first general depiction of local natural resources connected to the wider balance of the Earth's resources and climate. Von Humboldt decried the 'imprudent' activities of humanity, in Europe and in the Americas, depaupering the soils (and the general climate) by cutting down the forests and denounced the damage to natural resources as "incalculable" and possibly catastrophic if humanity continued to 'disturb the world so brutally' [18]. The conservation of resources was already a known concept in Europe in particular, with specific reference to the conservation of timber: in 1664, English gardener John Evelyn had written a book on forestry in which he clearly stated "We had better be without gold than without timber" [19]. A few years later Jean-Baptiste Colbert, France's Minister of Finance, will create the first national program to plant trees, without which, he said "France will perish for the want of woods" [20]. The intellectual context for sustainability was therefore already present in Europe of late XVII century, but it will take Humboldt's Naturgemälde and the later theorization of ecology to make it a pivotal concept in our understanding of the need for preserving natural resources in order to maintain humanity's balance with the natural environment [18].

Rural development was explored in 2003 by Marsden and in his book The Condition of Rural Sustainability, in which he presents three models of rural development dynamics. The first is an agro-industrial model, which was in use in Europe until the end of the 1980s; this is followed the post-productivist model which emerged towards the end of the twentieth century. Through this approach, rural areas are seen as a place of consumption. Its focus was on rural industrialization development and was mainly present in regions near big cities in northwest Europe. The third sustainable rural development model represents a broader approach including three aspects: economic, environmental and social. Its development has been supported by some European policies, especially by the Agenda 2000 of Common Agricultural Policy (CAP) reform [21].

By the end of 2010 and the development of Europe 2020 strategy, the "Smart Village" concept emerged [22]. Although the concept of "Smart Village", compared to the "Smart City" was way less present and discussed in the literature, in recent years this has changed [3]. Rural Development: Principles, Policies and Management by Katar Singh is dealing with all aspects of rural development addressing the basic idea, concept and elements, policy instruments, strategies, programs, and management of rural development [23].

In 2013, Green in his Handbook of Rural Development started to raise awareness of about the importance of villages and their future development in both developed and developing countries [24]. The concept of "Smart Village" was also explored by van Gevelt and Holmes. They proposed a vision for rural improvement by the development of different sectors such as health, education, food security, environment, quality of life [25]. Rural Development: Knowledge & Expertise in Governance (2015) by Kristof Van Assche provides an understanding about rural policies of how they have been implemented until then and what could have been done in the future [26].

In Sustainable Smart Cities and Smart Villages Research, the authors Visvizi and Lytras (2018) provide an overview of different socio-economic issues and aspects for both Smart Cities and Smart Villages, addressing community level problems. The book itself covers areas such as policy design, strategy development, case studies, technology-related issues, application tools and systems [27]. Another of their books, *Smart Villages in the EU and Beyond,* is built on the fact of rural depopulation, mainly focusing on EU villages, and

explores this problem and provides a conceptual framework for solving it with the use of ICTs [14].

In 2018, Zavratnik et al. provided a review of existing EU initiatives and practices in relation to the Smart Village concept, including the Slovenian example of Fab Village [15]. Two years later, Patnaik, Sen and Mahmoud in Smart Village Technology: Concepts and Developments explore different approaches for sustainable Smart Villages development, presenting current examples of how Big data and Internet-of-Things solutions contribute to it. Starting with SV policy overview and its relation to technology, it provides agricultural and water management, then renewable electricity management concepts and ends with an example of applying smart technology for problem solving in rural areas [28]. In the same manner, Cvar et al. showed how the use of IoT solutions in Smart Villages can lead to the improvement of rural inhabitants' lives [29]. Zavratnik et al. are more focused on people living in this area. They put the focus on communities themselves and the people living in these communities, suggesting that technology serves people and not the other way around [30].

### 3. Rural Policies Funding

The United Nations Sustainable Development (UN SDGs) Agenda 2030 consists of 17 sustainable people-centered goals covering different aspects of life [31]. It streams towards improving people's lives by focusing on access to quality food, health, education, electricity, industrialization, reduction in gender inequalities, climate-related actions, land protection and agriculture, sustainable consumption and production, sustainable human settlements (transforming our world: the 2030 Agenda for Sustainable Development) [32]. Not all of them have been developed at the same level. Some of them, especially that of sustainable energy development and access for all citizens, have been developing slower than expected [33].

In the context of rural development and in order to reach these goals, a number of rural-related policies have been formulated, supported by different funding programs. The European Regional and Development Fund (ERDF) together with the Cohesion Fund (CF) are the main sources of funding for regional policy. Another important funding source for these policies is the European Structural and Investment Funds (ESI) which consist of the European Social Fund (ESF), European Agricultural Fund for Rural Development (EAFRD) and the European Maritime and Fisheries Fund (EMFF) [34].

The EU rural development policy (EAFRD), together with the European Agricultural Guarantee Fund (EAGF) are funding instruments of the Common Agricultural Policy (CAP), the most well known and most important rural-related policy (rural development). The key EU funding instrument for supporting energy, transport and digitally based services such as telecommunication is the so-called Connecting Europe Facility (CEF) [35].

Connecting the Europe Broadband Fund (CEBF) as an initiative by the EC and the European Investment Bank (EIB) is supporting the rural network implementation [36]. Another financing instrument for rural high-speed broadband is the European Fund for Strategic Investment (EFSI) which also covers funding for other rural areas (education, health, energy, environment, social projects) [37].

There has been considerable development in the application of the concept of Smart Villages in Europe. Countries that engage with this concept started implementing their actions into practices. Interesting examples can be found in a number of countries, such as Slovenia [15], Czech Republic [8], Germany [38] France [39], Finland [40], Italy [41], as well as globally such as in Indonesia [42], Korea [43], Ghana [44], India [45], East Africa [46].

There is not a unique Smart Village model that applies for each country and region that can be developed and broadly implemented. Local or regional areas/communities can be very diverse and face different obstacles, even within the same country. However, the basics such as sustainable development goals are established as a global framework for future sustainable development. The Paris Agreement and the Agenda 2030 are future-

oriented global initiatives that cover all three aspects of sustainable development: economic, environmental and societal [47].

## 4. Research Design and Methodology

The purpose of this study is to contribute towards enriching the academic knowledge in this field, systematize all the policies and focus on their study and analysis. It can be beneficial for policymakers and future policy development. For the methodological part of this research, the following methods were used:

1.  Literature review of current SV policies for identifying categories of interest.
2.  Data collection through the literature review and online research with international stakeholders, using an online questionnaire.

The online questionnaire was distributed among the organizations which were part of the Alpine space region and the project Smart Digital Transformation of Villages in the Alpine Space (Smart Villages), co-funded by the Interreg Alpine Space program. In the project, thirteen partners were included from six different countries: Austria, France, Germany, Italy, Slovenia and Switzerland. The survey was distributed online. Respondents were able to answer it during the months from July to October 2020. The participants were from six EU countries and 11 Alpine space regions—Austria: Lower Austria, Upper Austria and Tyrol; France: Provence-Alpes-Côte d'Azur/Auvergne-Rhône-Alpes; Germany: Lake of Constance, Region Südlicher Obrrhein; Italy: Liguria and Area Metropolitana di Genova; Slovenia: Podravje; Switzerland: Region Luzern West and Upper Valais. In total, thirteen questionnaires were distributed among all project partners. A summary of these data can be found in Table A1, Appendix A.

The questionnaire consists of three main groups of questions. The first group is related to general questions for the respondent such as contact details, state and region, type of partners. They consisted of both closed-ended questions, where a set of responses were provided so that the respondent could choose from them, in addition to open ended questions.

The second group of questions were related to the existing policies of Smart Villages. They are divided by six different areas/pillars. Each question related to any specific area consists of six sub-questions: (a) the name/s of the policy instrument; (b) the coverage level of the policy; (c) responsible body; (d) link to the policy instrument; (e) sub-question related to the policies' positive impact and suggestion for improvements; (f) sub-question about what makes the policy successful. All sub-questions were open ended, except the question for policy coverage level, with possible answers ranging from the municipal to the regional and the national. As a last question from this group, they were asked to add a policy that covered any other area not mentioned previously.

Following the same structure from the second group of questions, the third group of questions covered future-related (2020+) policies of Smart Villages. In the same manner, the questions were divided by the area categories, consisting of the same sub-questions. In the data presentation part, the answers were presented by category following an alphabetical order of countries.

## 5. Rural Policies Categorization and Review

### 5.1. Rural Policies Categorisation and Research Area Description

In order for the Smart Village concept to be developed, good policies for rural development should be established. These can vary between regional, national and EU policies [48]. The OECD states that the vision for rural development and succeeding in the concept of Smart Villages should be a people-centered approach covering different but interconnected fields. The same source provides a list of rural areas that should be addressed separately [13]. Their categorization is in line with the one provided by the EU-funded project SHERPA [12]. Following both sources and their categorization, we identified six different areas related to rural development, to be of interest for our research.

The first and probably the biggest rural problem is connected to depopulation and population aging in rural areas. Rapid urbanization and the focus of smart cities development and increased opportunities made the rural–urban inequalities more visible, and thus cities more attractive than villages. This has led to villages' depopulation and migration to cities [27].

Global focus on climate change and the dependence on natural resources especially in rural areas led to new rules and policy development. That is why the area of climate change and environmental services is explored separately, as a separate topic.

The next topic is related to change in the production and diversification of the rural economy. Agriculture development follows rural development and in this context, it represents agricultural development through improved communication processes starting with the planning, design and development of new ways of ICT application in this domain [6].

It is natural that rural areas were connected to agricultural activities, whereas non-agricultural activities were connected to urban places, but with the growing interest in rural development, the non-farm rural economy has also increased [49]. However, there are still differences between these markets which can be classified into three categories: 1. physical distance from the main markets, which increases the cost of transportation; 2. lower competitiveness due to the homogeneity of production; and 3. limited growth opportunities determined by local conditions [13]. Local production is limited to the local markets and very often products are sold by very low prices either directly or through intermediaries [50]. However, the digitalization and the rapid growth of new technology allow easy information sharing and provide new possibilities for business [47].

Infrastructure and basic services are another topic that we give special importance to. Infrastructure and basic services such as access to clean drinking water, sanitation and waste management contribute to rural–urban gaps, as many rural areas fall behind providing these services to its inhabitants [47]. Electricity, more specifically renewable energy sources, are seen as an essential need that support the development of other rural services [33]. Smart Villages policies also strive for an overall improved management for providing sustainable renewable energy access to rural areas worldwide [2].

The next focus of this study is that of digital transformation and bridging the rural–urban gap. As the concept of Smart Villages is supported by the EU to a great extent, it is largely based on technology and more precisely on the use of the internet in facilitating the everyday life of people [4]. The European Network for Rural Development (ENRD) that connects rural development stakeholders and facilitates the process of knowledge and information sharing among EU member states also highlights the role of technology for future improvements. Digitization, or the process of switching from an analogue to digital form of information, together with digitalization, have seen technologically managed business processes affect economic and societal aspects of people's lives [51].

The main idea behind the development of Smart Villages is the collective well-being and improvement of social equality among rural and urban areas [52]. That is why the next topic is related to the social aspect of living. EU support for Smart Villages is coordinating different rural policies that together contribute to changing and enhancing people's lives [53]. Cultural dimensions, quality of life and wellbeing in rural areas is heavily dependent on rural development [54].

### 5.2. Rural Policies Literature Review and Survey Data Collection

Based on the described categories, the next section covers all of the policies, projects or actions for each area. The first group is systematizing all policies found in the literature review, whereas the second group presents all of the policies collected through the distributed questionnaire, including both current and future policies.

5.2.1. Existing and Future (2020+) Policies from Literature Review

Depopulation and Population Aging

Horizon 2020 was developed as a program for research and innovation including a range of elements for rural development [55]. Addressing this issue of depopulation, together with land abandonment and the loss of biodiversity, a Horizon 2020 project —Polirural—was established. Combining these three aspects, it provides a framework for the future development of rural areas and people [56]. Other projects such as Digitisation: Economic and Social Impacts on Rural Areas (DESIRA) [57], SHERPA [12] and Living Lab research concept in Rural Areas (LIVERUR) [58] are also part of Horizon 2020. They are included in this review, but placed in different subcategories, based on the areas they cover. Future policies regarding depopulation and population ageing can be found in the last category "Other", as policies that cover depopulation cover other areas as well.

Climate Change and Environmental Services

The Paris Climate agreement is a global agreement adopted in 2015, by around 190 parties. The key focus on this agreement was the reduction in greenhouse gas emissions and focusing more on renewable sources and energy efficiency [59]. Another policy is the Clean Air Policy Package which strives for increased air quality and pollution reduction [60].

Future-oriented climate and energy-related goals are set by the 2030 climate and energy framework, introduced in September 2020 by the European Commission as part of the European Green Deal. The policy consists of a few goals: reduction in gas emissions, more efficient use of energy, switching to renewable energy, reforms in the Emissions Trading system, competitive and integrated internal energy market, widely affordable energy and secure supply [61].

Change in Production and Diversification of the Rural Economy

One of the most important EU rural policies strongly affecting rural economy and people is the Common Agricultural Policy (CAP) [62]. It was launched in 1962, with the aim of supporting agriculture and farmers, supporting the rural economy and communities with employment opportunities, protecting nature by the control of natural resources and ensuring a sustainable and environmentally safe food production [63]. A similar policy is the Common Fisheries Policy (CFP) which has supported EU fishermen's and the fishing industry since the 1970s [64].

European Innovation Partnership for Agricultural productivity and Sustainability (EIP-AGRI) was launched in 2012 as part of the Europe 2020 strategy for rural development [65]. It aimed to accelerate the process of policy implementation by different activities such as knowledge sharing and common practices [66]. Similarly, the Smart Specialization Platform for Agri-Food (S3P Agri-Food) is a platform for cooperation among EU regions and their mutual development [67].

LIVERUR, as a Horizon 2020 project, is more focused on developing new business models and boosting local productions [58].

In regard to future-oriented policies, CAP- post 2020 sets its future policy objectives related to increased productivity by technological development and digitalization, increasing agriculture earnings and the establishment of a fair living standard, control over product supply and product prices and improve business opportunities in rural areas, attracting more young people [68].

Infrastructure and Basic Services

The International Renewable Energy Agency (IRENA) eases the process of connecting all concerned parties regarding this issue and provides an encompassing database of this issue for each country [69]. The pilot Project 'Smart Rural Transport Areas' (SMARTA) is a two-year project that aims to analyze the existing infrastructure and propose an improved

long-term model for the connection and improvement of rural transportation services [70]. No future policies are provided about this area.

Digital Transformation and Bridging the Rural–Urban Gap

Information and Communication Technology Policy covers aspects such as the internet, telecommunication and broadcast. ICT policy helps in providing all communities with digital technology, in this way allowing them access to information [71]. Another digitally related policy is Europe's Internet of Things policy, including policy actions such as the Alliance for Internet of Things Innovation [72] and the Digital Single Market Strategy [73] for the development of IoT in rural areas [74]. The action plan for rural broadband by the EC sets the objectives of implementing broadband connection in rural areas, improving it in others and developing a methodology for monitoring broadband investments [75]. Broadband Competence Offices Policies (BCO) support EU digital objectives and especially emphasize the importance of equal access to the internet in rural areas [76]. Free Wi-Fi for Europeans is a program for free wireless internet access for rural areas across Europe. The goal of this program is to allow the most centralized parts of the villages/cities to be connected to the internet [77].

EU Digital Single Market [78] and Gigabit Society [79], Broadband programming 2021–2027 and 5G are part of the future activities for the digital EU. The main goals include connectivity to all important socio-economic drivers in all European households and 5G coverage in urban areas, by 2015 [80].

The Social Aspect of Living

International Association Rurality-Environment-Development (RED) established a proposal for rural development post-2020, where rural areas should be developed and also integrated with the cities for mutual cooperation and cohesion. It is based on three base principles for future rural development: (1) rural–urban equilibrium that will allow strengthening and balancing this relationship; (2) citizens' rights equality which means that every citizen, as an individual, should be included in the development process in the same manner as an entity; (3) equality in the implementation of methods for territorial development projects [81].

Other

All of the above policies are covering one specific area. However, there are types of policies which are broader and more general in nature. These types of policies cover different fields together and are related to the regulation of several areas at once. We summarize them under one category "Other". Such a policy is the Rural Development Policy 2014–2020 as an EU policy for the regions. It covers environmental, economic and societal areas. The OECD rural development policy aims for the general improvement of the living conditions in rural settlements [13].

European Cohesion Policy is another growth policy that can vary between the local, regional and national levels and it is dedicated to both rural and urban development [82]. With the support of the European Commission's technical assistance contractor, AEIDL., the Liaison Entre Actions de Développement de l'Économie Rurale (LEADER), meaning 'Links between the rural economy and development actions' approach was developed. It represents a local development method, using local resources and citizen encouragement for people involvement in local areas development by formulating local action groups (LAGs) [83]. Community-Led Local Development (CLLD) is a similar policy that has been developed as an extension from the LEADER.

Cork 2.0 declaration is another integrated rural policy oriented towards rural prosperity, and changes in business models—value chains, preserving rural natural resources and other climate-related actions, invest in innovations and the general success of villages [84].

Focused on the economic aspect, sustainable development and enhancing citizens' lives, EU regional policy was developed as the EU's main investment policy [34]. Bled

Declaration for a "Smarter Future of the Rural Areas in EU from 2018 [48] and Social Innovation in Marginalized Rural Areas" (SIMRA)—Horizon 2020 project [85]—are also part of this category, focusing more on social innovation. Linking Actors, Instruments and Policies through Networks (LIAISON), Replicable Business Models for Modern Rural Economies (RUBIZMO), Rural–Urban Outlooks: Unlocking Synergies (ROBUST), Social Entrepreneurship in Structurally Weak Rural Regions: Analysing Innovative Troubleshooters in Action (RURACTION) are similar projects [14]. The European Spatial Planning Observation Network (ESPON) 2020 Cooperation Programme is a strategy for rural development, developed by the EU for rural cohesion, which places great focus on the depopulation problem.

Focusing more on the environmental-related issues we are facing today, the European Green Deal development strategy was established. It is part of the United Nations 2030 agenda and strives for a sustainable future and an improvement in future generations' lives. The European Green Deal consists of varying future-oriented policies, covering different aspects of everyday life, based on the UN sustainable goals. The Green Deal contains eight elements: EU's climate ambition for 2030 and 2050, meaning that the EC is determined to implement more rigorous climate actions; prioritizing energy supply and efficient use and secure energy, based on renewable sources; mobilizing the industry, focusing on sustainable product choices and promoting a circular economy; a more efficient way of building and construction; shift to smarter logistics and transportation; focus on safe and quality food; action plan for preserving biodiversity and actions for pollution prevention [73].

Sustainable Hub to Engage into Rural Policies with Actors (SHERPA) is a project part of the Horizon 2021–2027 program which supports rural policy development [12]. Digitisation: Economic and Social Impacts on Rural Areas (DESIRA) is another Horizon project focused on social transformation in rural areas [57].

5.2.2. Existing and Future (2020+) Policies from Survey Data Collection
Depopulation and Population Aging

In the Austrian region of Tyrol, the depopulation and migration from rural areas is addressed by a regional development policy, led by the Federal Ministry for Sustainability and Tourism. The policy's original name is Zukunftsraum and its concept describes that improvement in tourism only is not a solution itself for depopulation and the focus should be put on people and matters like gender equality and job inclusion. This policy was found to be very successful in raising awareness of rural areas in Austria [85]. Another policy from Austria was in the region Upper Austria. Its name is Europäischen Fonds für regionale Entwicklung (EFRE). The coverage level of this policy is European with regional implementation and is developed by Land Oberoesterreich. The policy is mainly helping companies and the goal is reaching equality between regions. More information about this policy can be found in [86].

In France, the country addressed the consequences from population aging in rural areas through changes in the law legislation, encountering the requirements of older people. The Act on adapting society to an ageing population was put in force. Anticipate, adapt and support are the three postulates on which the changes were done. The law for society adaptation to an ageing population covered all regions and was developed at a national level. This created an autonomy financial allocation for ageing people, and different measures around home care for elderly people, transport and other social related activities [87].

The Platform for Neighborhood Help and Onboarding of Newcomers was developed in Germany at a municipal level by Bodensee StandortMarketing, EDG, City of Arbon. The city of Arbon is located on the Lake of Constance, or Bodensee in German language, which borders four countries: Germany, Austria, Switzerland and Liechtenstein. This is a policy that focuses on people and aims to integrate newcomers into the community. Neighborhood aid is striving towards promoting cohesion in rural areas, by developing neighboring communities [88]. Entwicklungsprogramm Ländlicher Raum (ELR) is a municipal policy

in Germany in the region Südlicher Obrrhein which mainly focuses on rural demographic development, nature and land protection, offering an integrated, systematic rural area development. The responsible body for this policy is the state government. More details can be found in [89].

The issue of depopulation and population ageing in rural areas in Italy has been approached by a few policies. The first one is the National Strategy for Internal Areas (SNAI), a national policy instrument that works on groups of municipalities, including a total of 1077 municipalities in the whole country. The responsible body for SNAI at the regional level is Regione Liguria Sviluppo Economico. The policy works towards a territorial cohesion that deals with the problem of demographic decline, which is a typical issue of this country. It is characterized by the active role played by the municipalities and Park Authorities and their involvement in the definition of the strategy and actions. The planned investments derive from the shared definition of a common strategy and a plan for monitoring the implementation of the planned interventions. The effectiveness of the policy for internal areas can be improved through the further development of a territorial marketing portal, which enhances the present potential [90]. Another policy in Italy is Measure 19 RDP Regione Liguria Agricoltura. It is a regional instrument, but it works at the municipal level through Local Action Groups (LAGs). The responsible development body is Regione Liguria Agricoltura, as provided in [91]. The following policy is The National Operational Program Metropolitan Cities 2014–2020 PON METRO. The responsible body is the Agency for Territorial Cohesion (it. Agenzia per la Coesione e Città metropolitan) [92]. All of the above policies in Italy are in progress and respond to this concrete rural problem.

Age Policy named Altersleitbild Kanton Luzern was developed in Switzerland, Region Luzern West. The responsible development body is the Kanton Luzern. It deals with demographic development, and the improvement of general living conditions for elderly people [93].

In regard to the future-oriented policies, the Natural Regional Park of Vercors Chart in France is a future-oriented strategy about depopulation. This policy is being created via different workshops, conducted at a small local level with the local participants. The chart should be all oriented to sustainable policies [94].

The Italian strategy of the Rural Development Plan of the Liguria Region (PSR 2021–2027), developed at a regional level by the Liguria Region, for the new Rural Development Plan, is under construction by the competent bodies. To this date, there are still no documents related to the new programming. This is an experiment of collaboration between local action groups (LAGs), created to contribute to the debate on the local development of rural areas and to promote the participatory discussion of local development issues, in the framework of the implementation of the 2014–2020 strategies and in the perspective of the European programming 2021–2027. The policy instrument can be improved by more effectively adopting the core principles of the LEADER approach. What makes the policy useful is the possibility to listen and work with local communities [95].

In Slovenia, in light of caring for the elderly, the respondent identified the policy of Caring for the Elderly and New Opportunities for Rural Employment, run by the Municipality of Kungota and under the supervision of Sociolab and Prizma Foundation. For the time being, the positive impacts are still being generated, although the policymakers are actively working on these activities. The key added value that made these activities successful is the multifaceted nature of networking and collaboration [96].

Climate Change and Environmental Services

In Upper Austria, Klimaschutz in Oberösterreich policy is present at a regional level, developed by Land Oberoestrreich. It promotes the use of renewable energy sources, as stated in [97].

In France, in the region of Provence-Alpes-Côte d'Azur, a regional Scheme about Climate, Air and Energy was established. This regional level policy works for developing a broad climate change alteration plan, including local, more generic schemes such as

territorial climate energy plans (PCET) and territorial coherence schemes (SCOT) about energy and territory transition. They are more operational tools developed to change territories with concrete actions [98].

Integrated Energy and Climate Protection Concept (IEKK) (German: Integriertes Energie- und Klimaschutzkonzept) is a climate-related German policy developed at a regional level. It strives for greenhouse gasses reduction, regulates energy protection, including different areas of activity such as electricity, heat, traffic, land use and material flows [99].

In Italy, there are four climate-related strategies provided by respondents. The strategy for Sustainable Development was developed at a regional level by Regione Liguria, with a key characteristic of the involvement of local institutions and stakeholders. Information and territorial animation activities are in relation to the process of construction of the regional strategy for sustainable development [100]. Rural Development Plan Measure 5 was also developed at a regional level by Regione Liguria under this strategy, and regions are preparing their own sustainable development strategy and appropriate objectives [91]. The third policy, the Metropolitan Strategic Plan (MSP), was developed at the metropolitan area level in relation to climate change issues. The responsible body for this policy is the Metropolitan City of Genoa. This policy represents a participatory process that has actively involved all municipalities and other stakeholders and creating strategies. The strategic lines have been articulated on the territorial, economic and social, institutional dimensions, focusing on coordinating change, developing the Genoa metropolis, optimizing services, adapting to climate change, building the sense of belonging to the Metropolitan City. The first MSP (approved in 2017) mainly directed towards its policy actions by the metropolitan administration and to rethink its organizational model according to the strategies. However, the update of the MSP will have to open up its policies to further public and private actors and develop relations with the different levels of [101]. Another policy from Italy is Proterina3Evolution (Maritime IT–FR Program) developed between Italy and the France Maritime area (Proterina). The responsible development body is the LeadPartner: Centro Internazionale di Ricerca in Monitoraggio Ambientale CIMA Research Foundation (Proterina). This policy creates communities resilient to the negative effects of climate change, the activation of forms of active participation of the population and institutional stakeholders. The project has a positive impact for the contents of innovation that it has brought, both for the planning aspects and for the implementation and monitoring of experimental intervention. Further improvements can be achieved through the application of the results in other territorial contexts [102].

The Slovenian respondent identified Nacionalni energetski in podnebni načrt which is an action strategy paper setting out the objectives, policies and actions on the five dimensions of the Energy Union for the period up to 2030. As the action strategy has been recently activated, the results are yet to be produced, but the respondent stresses the importance of the participation of the local communities [103].

Cantonal Energy Law Luzern Large Consumer Model—(original name: Umweltbericht Kanton Luzern, Energiegesetz Kanton Luzern, Bericht in Erarbeitung) in Kanton Luzern, Switzerland, focuses on five environmental challenges for this region that the canton is facing and tries to provide a response to them. The report formulates environmental goals for each area that are to be achieved by 2030—across topics, in cooperation with those responsible and other actors [104].

Climate change and environmental services are of great interest and the regulation for this can be seen in many policies that follow. Lower Austria Climate and Energy Roadmap 2030 is related to major shifts connected to electricity, the use of renewable and long-term efficient energy sources, forbidding coal and oil use for heating and transportation [105]. The Tirol 2050 regional policy is a future-oriented strategy related to climate actions in Tyrol, Austria. This deals with climate-related problems in communication and involving local inhabitants in action. Its goal is to make the region of Tyrol independent regarding the supply of energy for this region and its inhabitants [106].

In France, an intern municipality policy has been developed, named the Ecologic Transition Contract. This contract is an operational document including funded actions at a local scale. It was built with a large panel of citizens in the context of a climate emergency. This is future oriented in the period of the next three years [107].

In Germany, Region Südlicher Obrrhein, a state government regional policy—the Integrated Energy and Climate Protection Concept (IEKK)—is a policy related to this topic. It sets clear goals for future actions, including electricity, heat, transportation, land and material use in their focus [99].

In Italy, a future strategy under construction is the Sustainable Development Strategy (it. Strategia Regionale per lo Sviluppo Sostenibile) in the region Liguria. The strategy aims to apply the indications of Agenda 2030 at local level. The added value was that of involving local institutions and stakeholders to construct a strategy and define the objectives of sustainable development at the local level [95].

Regarding future policies in connection to climate change and environmental services, the Slovenian respondent reported the adoption of local ordinances and restrictive measures, implementing at the national level and with the responsible bodies of ministries and municipalities. However, a concrete policy instrument remains to be designed. In these terms, positive impacts and added value, generated by the policy instruments, cannot be reported.

The policy Klimabericht in Erarbeitung was developed in Kanton Luzrn, Switzerland, to increase awareness on climate-related topics [104]. Change in the production and diversification of the rural economy.

In Tyrol, Austria there is the Tiroler Forschungs und Innovationsstrategie policy. It is related to economic development and creates community support that aims to create an innovative economy and increase in region attractiveness, by boosting tourism based on new innovative projects [108]. Another policy in Upper Austria is Rural Development, or originally called Ländliche Entwicklung, developed by Land Oberosterreich at a regional coverage level. More about this policy is provided in [109].

In France, the development of rural areas is to a great extent regulated by the French legal framework. Some measures from that law allow specific measures and funding for rural areas about economic development. A good characteristic about this regulation is the creation of new dynamic rural areas with special measures about social disposals and financial taxes [110].

BodenseeMittelstand (BoMi) 4.0 is a regional policy by the University of Technology, Business and Design Konstanz in Germany, or the region of Lake of Constance. The policy represents support for small and medium-sized enterprises from rural areas which do not perceive digitization as a "risk" but as an "opportunity". To this end, the project supports small and medium-sized enterprises (SMEs) from Germany, Austria, Switzerland and the Principality of Liechtenstein in their digital transformation by bundling and coordinating the expertise from business, science and SME-related institutions available around the lake and making it more accessible to regional SMEs [111]. Top in the country! Technology Leader for Baden-Württemberg (Spitze auf dem Land) is a policy developed by companies in Germany, Region Südlicher Obrrhein, with the state government as a responsible body. It is part of the Rural Areas Development Program (ELR) and it is oriented towards developing and promoting innovation and circular economy with innovative companies [112]. The policy is also future-oriented towards improving the development of the rural economy [89].

The Rural Development Plan developed by Regione Liguria in Italy is working on improvement of the environment in the countryside and the possibility of sustaining investment in rural development and the agricultural economic sector [92]. The Italian policy Rural Development Plan of the Liguria Region (PSR 2014–2020), Growth Act—Regional Law n.1/2016, was developed at the regional level by the Liguria Region. The key policy feature was the construction of the plan focused on a major theme—the smart, sustainable and inclusive growth of rural areas. An important aspect was that each program proposed

different actions based on the needs of the sector which fill the structural weakness of a fragile sector with specific interventions. Some measures in the Plan provide incentives for the rural economy and productive diversification. The measures had a positive impact on providing economic support to companies [113]. Based on this policy, the Rural Development Plan 2021–2027 was formulated. Its postulate was that knowledge and innovation were determining factors to be the driving force of rural development processes. The strategy for the new Rural Development Plan was under construction by the competent bodies. To this date, there are still no documents related to the new programming. The policy instrument can improve by more effectively adopting the core principles of the LEADER approach. The possibility to listen and work with local communities [95].

In Slovenia, there is an established a Program razvoja podeželja Republike Slovenije 2014–2020 (Rural Development Programme 2014–2020) with the Ministry of Agriculture, Forestry and Food as a responsible body. The identified program was established at a national level and funded action under five out of six rural development priorities. The priorities are the following: 1. priority on "knowledge transfer and innovation in agriculture, forestry and rural areas"; 2. priority on "competitiveness of agri-sector and sustainable forestry"; 3. priority on "food chain organisation, including processing and marketing of agricultural products and animal welfare"; 4. priority on "restoring, preserving and enhancing ecosystems related to agriculture and forestry" and priority on "local development and job creation in rural areas". According to the Slovenian respondent, for the time being, the identified policy has not yet been sufficient. On the other hand, it has been rather successful in enabling the increased accessibility and transparency of aid measures [114].

Future policies regarding change in the production and diversification of the rural economy are—according to a Slovenian respondent—addressed in the Skupna kmetijska politika (Common Agriculture Policy) 2021–2027 with three general objectives of promoting food security, strengthening environmental concerns and climate action and rural areas as well. The policy instrument is covered by the Ministry of Agriculture, Forestry and Food, however, its positive effect remains to be seen [115].

As provided in [91], Kantonaler Richtplan is a policy that will cover this issue in the post-2020 period, in Kanton of Luzern, Switzerland.

Infrastructure and Basic Services

Eco plus, The Business Agency of Lower Austria, is an Austrian agency from this region that helps businesses invest in different kinds of transportation infrastructure, including improvements in digital infrastructure. This policy improves business-to-consumer relationships, providing wider market access for companies, as described in [116].

In France, the Departmental Scheme for a Better Accessibility to Services developed by the Department council and local state gathers all stakeholders from basic services together in order to discuss the real needs of the territories. These include different sectors such as health, basic goods, transports and digital services on one side, and users on the other, in addition to how their access and cooperation can be improved in rural areas [117].

The digitization of administrative processes in the city of Tengen is the policy in Germany, the region of the Lake of Constance, developed at a municipal level by the City of Tengen and Bodensee Standort Marketing GmbH. The policy is digitizing the process of ordering or changing the waste bins in the waste management system. By involving the citizens in the process, the platform has been widely supported [118].

In Italy, the previously mentioned National Strategy for Inner Areas (SNAI), as a national policy instrument that works among groups of municipalities, is also dealing with this area of infrastructure and basic services. The added value of this policy is to plan specific interventions to meet the needs of the territory [119]. Another policy also mentioned in the depopulation and population aging context in Italy is the Measure 19 Rural Development Plan. It is a regional instrument, which works at the municipal level and provides regulations about infrastructure and basic services in rural areas [91]. The basic rural services are regulated by some other policies such as the Regional Social and

Health Plan, Metropolitan Strategic Plan (MSP), Territorial Plans (PTR, Metropolitan PT), Sustainable Mobility Plans (PUMS), Area Plan (Optimal Territorial Area—Water and Waste ATO) and Municipal Urban Plans (PUC). They can vary between the municipal, regional and national levels [101].

In regard to infrastructure and basic services, the Slovenian respondent did not name any specific policies, but on the other hand, has identified the levels where policy measures in the form of documents and accompanying strategies were taking place. In this respect, the respondent specified national, regional and local level policy measures. In terms of the local level, the respondent reported regular municipal activities and added that they are beneficial as the local policy makers know best what is good for their local environments, especially in terms of the challenges these areas are facing. Otherwise, the success of these activities also comes from the relatively small size of the municipality of Kungota, which runs the policy measures, resulting in greater accessibility and citizens' participation [120]. In the same manner, the respondent outlined a collection of municipality's development documents, run by the Kungota municipality, concretely annual budget. According to the respondents, the budget has had a positive impact, also due to the fact that it was formed following a sustainable position of the proposed budget [120].

A national Swiss policy developed by the Swiss confederation proposes a report of the federal council on services of general interest. This is the first overview of public service in Switzerland in the areas of transport, postal services, radio and television, telecommunications and energy supply. The inventory is supplemented by a review of the current situation following the reforms of recent years, as well as an analysis of the challenges and guidelines for future basic service policy. The report provides not only the legislator but also all citizens with a basis for forming opinions on the future design of the basic infrastructure service. By "public service" the Federal Council understands a politically defined basic provision of infrastructure goods and services, which should be available to all sections of the population and regions of the country according to the same principles, in good quality and at reasonable prices. In other words, the scope of the basic service is determined by the legislature. Public services must be continuously available in all parts of the country, be affordable and meet the needs of the population and the economy. Quality is monitored and enforced by public authorities [121]. Mobilitätsstrategie in Erarbeitung is a mobility strategy developed at a regional level in Kanton Luzern, Switzerland. It covers topics such as global trends and technological changes, including new forms of mobility and public transport [122]. Another related policy from the same region is the ÖV-Berich, a public transport strategy that represents a public transport report of current and future possibilities of new transportation and how they should be implemented [123].

The aforementioned policy [89], Kantonaler Richtplan, is a policy that will deal with this problem of infrastructure and basic services in the post-2020 period, in Kanton of Luzern, Switzerland.

Digital Transformation and Bridging the Rural–Urban Gap

Regarding the next area of research, digital transformation and bridging the rural–urban gap, the policy for a digitalization strategy was developed in the lower Austria region with the goal of providing broadband internet to all citizens. The strategy tackles the related problems such as digital fit or ensuring that the population is interested in these issues by raising awareness, improving the population's digital competences and skills, allowing access to adequate digital infrastructure and offering digital solutions for business process improvements [124]. In Tyrol, Austria, the digital transformation and connectivity services are supported by the regional policy Breitband Masterplan 2019–2023 (Breitbandstrategie 2019–2023) with the main objective of fast broadband connection. Breitband Masterplan 2019–2023 is a regional policy related to basic services in the region of Tyrol, Austria [125]. A third strategy from Upper Austria, called Digital Region Upper Austria, was developed at a regional level with the responsible body of Land Oberosterreich [126].

In France, a policy called the Regional Coherence Strategy for Digital Development (SCORAN) in Provence-Alpes-Côte d'Azur (Fr: Stratégie de Cohérence Régionale d'Aménagement Numérique de Provence-Alpes-Côte d'Azur), was developed at a regional level. It allows a coherent deployment for fiber optic, and also a regulation for mobile networks. The infrastructure is provided based on the real needs of the area and it represents a legal obligation for territories and private operators to cover all the French territories (including rural "not valuable" areas) with fiber optics and mobile networks [127]. A French policy SCOT of the Grenoble Region—Territorial Coherence Scheme, is developed for a very specific coverage level of 800,000 habitants. It is a coherence scheme for the upcoming twenty years [128]. Standort Guide Bodensee is a regional policy developed by Bodensee Standort Marketing GmbH in Germany. The policy offers support for companies in their digitization process, trying to minimize the digital gap in rural and urban areas and make the rural areas more attractive for living and working [129]. Digitalakademie is a digital academy in Baden-Württemberg, Germany, that supports this region regarding its digitally related activities. It is developed at a municipal level and the responsible body is the state government [130].

In Italy, an Italian Banda Ultra Larga (BUL), meaning Ultra-Broadband strategy for ultra-wideband technology was developed at a regional level by the Regione Liguria. This represents a strategy for ultra-fast broadband connection for rural areas [131]. The Strategic Digital Program of the Liguria Region 2016–2018 is a policy in Italy, Area Metropolitana di Genova, developed by the Liguria Region. This strategy aims for achieving the faster implementation of digital infrastructure to overcome the "digital divide" [132].

In the Slovenian region at stake, the respondent has identified the policy measure of digitalization of the rural area, which is being implemented at the municipal level and run by the Municipality of Kungota with the partner organizations, with some of the funds coming from the Interreg Europe and Horizon 2020. To date, the activities have proven to be successful due to the direct involvement and networking with the stakeholders who report on the most pressing matters of their environment [133].

The "Digital Switzerland" policy strategy, developed by the federal government, is a broad strategy that coordinates different digital activities in Switzerland. It focuses on various priority areas for implementation, like the digital economy, data and digital content, as well as Switzerland's exchanges with other countries, with a particular focus on the digital domestic market of the EU. Despite all of the actions taken until now, it seems that the strategy is not yet well known and above all, implemented at a cantonal and regional level [134].

Another policy from Switzerland related to this area is Gesetz zur Förderung der digitalen Transformation in Graubünden. Its coverage level is also regional and the responsible body is Canton of Graubünden. It has initiated digital projects and represents a lean law tailored to close the digitalization gap between Graubünden and other more advanced cantons. The canton should set up a specialist committee to promote digital transformation and accelerate the digital transformation in line with the objectives. This body should pick up on those trends and topics that are relevant for Graubünden in connection with the digital transformation in exchange with the relevant actors from the various areas and sectors. The aim is to bring together innovative forces and digitization experts from various areas and sectors who will identify, initiate, accompany and coordinate digitization projects and support the canton in its tasks under the new law. The committee will also examine the feasibility and effectiveness of digitization projects and make recommendations to the canton on how to promote them. Finally, a contact point for industry organizations should be offered for questions concerning digital transformation [135]. In Switzerland, in the Kanton Luzern, the Planungsbericht Regionalentwicklung policy for the digital transformation activities has been designed. It is part of the regional development planning where the goal is to increase job opportunities in these areas in order to reduce the inequalities between the regions and keep a centralized settlement in the canton [136]. NRP—Projekt Wege zur Hoch Breitbandversorgung—is a future policy in Luzern, Switzerland, representing a

platform for a future rural development. More about the future development of this policy can be seen in [137].

The Social Aspect of Living

The NÖ Kulturwirtschaft (NÖKU), meaning Lower Austrian Cultural Industries Group is a cultural management organization in the lower Austria region composed of more than thirty institutions. The aim is to both press on with the promotion of individual institutional brands and clearly identify, establish, and systematically work on strategic content-based corporations, the development of formats, and fields of art. In close co-operation with the state of Lower Austria and its Department of Art and Culture, the NÖKU Group continually strives to develop and present artistic or scientific projects of thematic and social relevance with a cross-regional or international reach [138]. In Tyrol, Austria, the promotion of social inclusion and fight against poverty and discrimination is regulated by a policy ESF—Strategie Tirol 2020. It is being developed at a regional level by Land Tyrol [139]. In Upper Austria, the policy Upper Austrian Culture Quarter is being developed at a municipal level by the city hall. This policy is provided in [140].

The social aspect of living in Italy, Liguria, is covered by regional social and health plans. The responsible body for health and social policies is the Regional Health Authority in Liguria. It is a region in Italy that actively takes on this problem and has also established previous regulations [141]. In Italy, the Area Genoa, territorial marketing portal policy represents an important opportunity in the co-design and collaboration between institutions on the territory with the aim of increasing the awareness of its potential at the local, administrative and stakeholder levels. The Metropolitan City of Genoa wants to complete the section "empty to take", a mapping and filing of disused public buildings (former colonies, palaces, castles, fortifications) that insist on the entire territory and constitute a heritage to be discovered and enhanced, creating attractiveness, opportunities for young people, reducing land consumption and thus implementing environmental, social and economic sustainability [142].

In regard to the category of the social aspect of living in Slovenia, this particular respondent specified the so-called local plan for people's health and well-being, which takes place at the level of the municipality of Kungota and has had till now a positive impact on the community of the municipality. The added value of this particular set of activities is in the increased local population activity in the local environment [143]. Regarding the social aspect of living, the respondent from Slovenia put forward the data coming from the municipality level of Kungota. In this manner, the respondent identified the local action plans of the chosen municipality with the municipality also implementing the named policy measures. According to the respondent, the adopted measures are having a positive impact due to the involvement of the local community [120].

Other

In the next category "Other", we combined policies that covered at least three of the aforementioned areas under one policy. A general rural policy of this kind is the Austrian Rural Development Programme 2014–2020 which covers categories for rural development, such as the agricultural sector, regional economy and social development. It is supported by the national networking agency "Netzwerk Zukunftsraum Land LE 14-20" that facilitates connection among interested stakeholders [144].

The regional council in France developed the policy Regional Planning, Sustainable Development and Equality of Territories (SRADDET). It is an important policy that ushers designing other local coherent policies with a focus on sustainable development [145].

An Italian policy of this kind is that of the Metropolitan Strategic Plan (MSP). This covers all related areas to Smart Villages and rural development. A responsible body for development is the Metropolitan City of Genoa [101].

The policy in Switzerland, Federal Programme Digitalisation and Services of General Interest (Modellvorhaben Digitalisierung für die Grundversorgung nutzen) was developed at a national level by the Swiss federation, covering areas of smart mobility, smart living, smart economy, smart environment, smart governance and smart people. It combines services of general interest (SGI) with digitalization and makes a concrete attempt to exploit digitalization in a cross-sectoral field to a maximum [146]. Another Swiss national policy covering all the aforementioned areas is the law on the promotion of innovation; Innosuisse, Swiss Innovation Agency. This represents a framework and budget for the implementation of concrete projects. However, cooperation between innovation-agents and the respective cantonal services should be improved in order to increase the potential of synergies [147].

In Germany, a policy developed by the national government, Heimat 2.0, will be focused on establishing sustainable and stable communities, thus providing equal living standards [148].

A future rural policy in Italy, Metropolitan Agenda for Sustainable Development (Agenda 2030), comes as an update of the Metropolitan Strategic Plan, developed by Metropolitan City of Genoa. The agenda is based on a "bottom-up" approach by the active involvement of the municipalities, starting from the specificities and problems of the territories, the local landscape and the people who live there. A first experimentation of the "metropolitan charter of sustainable services" was carried out, which, thanks to defined rewards, encourages synergic actions oriented to sustainability (e.g., public transport, shared mobility, sports facilities, culture, spaces for co-working, etc.). The update of the metropolitan strategic plan is achieved through the construction path of the metropolitan agenda for sustainability, implementing the objectives of Agenda 2030, through collaboration and comparison with other Italian metropolitan cities (TO, MI, VE). The metropolitan agenda provides the definition and testing of a "model of sustainable urban space"; it is innovative as it integrates several lines of action (resilience, zero emissions, zero waste, soft mobility and integration with public transport, training and education, etc.), and it is also characterized by replicability and scalability to different realities and territorial dimensions (e.g., homogeneous area, urban areas, coastal areas, inland territories) [101].

New Regional Policy 2024+ and Application Programmes is a general policy developed by the federation/cantons in Switzerland that supports cantons and regions in the design and implementation of the application program. The policy is mainly active in regional centers and supports export-oriented companies as a priority. Consequently, the real peripheral settlements that work rather according to local, regional cycles are not beneficiaries of this policy. It could be further improved by emphasizing the importance of digitalization in all sectors and the competences of stakeholders in this field should be improved [149].

## 6. Summary of Research Findings

This section of research findings discussion follows the order or research goals stated at the beginning and provides an analysis of the results collected from both the literature review and online survey. Regarding the first research goal, establishing the rural areas and their analysis, we can provide the following information:

- From the literature, the greatest focus until now has been put on the area dealing with the digital transformation and bridging the rural–urban gap, in both existing and future-oriented policies, followed by change in the production and diversification of the rural economy.
- From all of the responses, we can see that the areas covered with existing policies are all approached with the same extent and importance. The greater focus is put on the depopulation and population aging issue. On the other hand, it can be noticed that slightly less present than all others are policies related to climate change and infrastructure and basic services.

- The focus of future policies is in the areas of climate change and environmental services. The least covered area in future policies is the social aspect of living, together with infrastructure and basic services.

  Related to the second research goal, we provided a discussion about policy characteristics:

- General timeline comparison of the policies indicates that future-related policies are less covered in this paper, which means that either participants are not aware of them or they do not exist.
- What is important for policy development is the coverage level. From the survey results, we can conclude that around half of the total policies—both existing and future—are developed at a regional level. From the rest of them, with a slight difference, national policies are more present than municipal ones. Other than this, we could see responses of policy development on an intermunicipal level, canton, sub-region areas.
- Municipally regulated areas are mainly those that regulate policies about the social aspect of living.
- Regionally regulated areas are climate change and environmental services and also digital transformation and bridging the rural–urban gap.
- Nationally regulated areas, together from existing and future policies, are mainly those related to depopulation and population aging and change in production and diversification of the rural economy.
- In terms of the social aspect of living in an area, policies are mainly developed at a municipal level. A very low number of policies for this area were developed at a national scale.
- Change in the production and diversification of the rural economy area has largely been regulated by regional and national scale policies.

  In line with the third research goal, we highlighted the main characteristics for each country explored in this paper:

- From the six countries included in this research, we can conclude that the country with the greatest number of policies related to rural development is Italy. Most of its existing policies are related to climate change and environmental services and most future policies are related to depopulation and population aging. The level of development of their policies are regional, but also metropolitan.
- After Italy, countries ordered by a decreasing number of policies are Switzerland, Austria, Germany and France.
- The country with the lowest number of policies is Slovenia. This can be explained by the fact that the development program was formulated for a specific region in this country and thus, in this online survey data analysis, only one region was included.

## 7. Policy Recommendations

The Smart Villages Project is a strategic implementation initiative of the EU-Strategy for the Alpine Region (EUSALP). The project is running from 2018 to 2021 and is financed by the Interreg Alpine Space Program. The policy recommendations coming out of this project are thus a major contribution to the implementation of EUSALP. The draft recommendations were elaborated in autumn 2020 by the project consortium. The draft recommendations were presented and discussed at the International Smart Villages policy conference on 10 November 2020 with 150 participants from all over Europe. Initially, this meeting was meant to be a physical meeting to be hold in Bern (CH). However, due to the COVID-19 pandemic, the meeting was reorganized into a virtual-only meeting. The draft recommendations were amended after the discussion at the International Smart Villages policy conference. They will flow into the policy cycle of the EUSALP and will be made available to policy makers in the Alpine area and outside of it like the other Macroregional strategies, the EU Commission, managing authorities and so on.

The Smart Villages approach is important for villages in mountain and rural areas to become more attractive and vibrant. Smart Villages are not only attractive for residents, but also for people from outside, who may stay in these villages for a certain period either as tourists or for work (concept of "third places" with e.g., coworking spaces). In this respect, the concept of Smart Villages helps also to create linkages between urban and rural areas. The Smart Villages approach helps those villages use the potential offered by digitalization and to bridge the natural handicaps of distances. With the Smart Villages approach, the communities can contribute to the European Green Deal and master their smart transformation. Finally, but not less importantly, the COVID-19 crisis has shown that Smart Villages are much more resilient to such a crisis.

The Smart Village approach is an integrative approach using the potential offered by digitalization and developing new solutions through participatory processes, thus relying on open and social innovation. This basic understanding of the Smart Village approach leads to the following policy recommendations. These policy recommendations address all institutional levels ranging from the EU level, through the macroregional level to national, regional and local level. Where appropriate, the respective level is directly addressed, and good examples are given.

R1: Consider the smart transformation of mountain, rural and peripheral villages as a strategic priority.

The smart transformation of mountain and rural villages helps bridge the natural handicaps of those areas and give them new perspective. In some mountain areas, with low population density, the Smart Villages concept can also help develop digital services and mobility offers for instance, while operators have been reluctant to invest in local infrastructure projects that are not viable. The smart transformation of those areas should therefore become a strategic priority. At the EU-level, a strong focus is already put on this topic with amongst others the EU action plan on Smart Villages, the activities carried out by the ENRD Network on Smart Villages and the new intergroup RUMRA & Smart Villages in the EU Parliament. This intergroup is an ideal platform to evaluate, whether it would be appropriate to create an explicit legal basis or an overarching strategy for the Smart Villages approach at the EU level. At the macroregional level (EUSALP), Smart Villages is considered one of the five strategic priority policy areas for the period 2020–2022. This is already a major success of the ongoing Smart Villages Interreg Alpine Space project. The creation of a network of Smart Villages in the Alpine area is envisaged in 2021. The seven countries and 48 regions represented in EUSALP are invited to actively support this process. At the national level, several countries have already integrated the Smart Villages approach into national strategies, such as the "Strategie Digitale Schweiz" in Switzerland and the Strategy for Inner Areas in Italy or the rural development program of Slovenia. At the regional level, the Smart Villages approach must also become a priority, such as, e.g., with the Law on digital transformation in Graubünden (CH). The same evidently holds true at the municipal level, as exemplified by the city of Tengen in Germany.

R2: Embed Smart Villages in existing and future strategies and policies.

The Smart Villages approach is an inter-sectorial approach. It covers many areas ranging from, e.g., tourism and mobility to e-government and to energy. It is therefore crucial, that the Smart Villages approach is embedded in existing and future policies. The Smart Villages concept should, e.g., be integrated into Pillar II of the CAP and in the cohesion policy (including cross border cooperation) and a certain budget allocated to it. The rural development programs during the period 2021–2027 including the LEADER/CLLD-approach should have a strong focus on digitalization and open/social/technical innovations. As regards the cohesion policy, the operational programs at national and regional level should also include special lines on the Smart Villages approach. From the side of the EU regulation, two out of the five policy objectives for 2021–2027 are offering significant potential for Smart Villages: Priority Objective 3 (a more connected Europe—mobility and regional ICT connectivity) and Priority Objective 5 (Europe closer to citizens—sustainable and integrated development of the urban, rural and coastal areas through local initiatives). Priority

Objective (PO) 3 addresses the more technical aspect, and PO 5 the community-based aspect. These opportunities need to be taken up in the operational programs at the national, regional and cross-border levels. At this actual stage (autumn 2020) when the programs are being drafted, it is therefore important, that stakeholders interested in the topic of Smart Villages contact their respective national and regional authorities. The Smart Villages approach is not only relevant for agricultural and cohesion policy, it is as well relevant for transport, education, health, social care, tourism, energy, housing, etc. All the relevant policies should therefore take the Smart Villages approach into account and encourage it.

R3: Allocate funds to integrative approaches such as the Smart Villages.

Integrative approaches such the Smart Villages approach face the common problem, that there is no dedicated funding available. Specific funding schemes should be established at all levels to allow such approaches to be developed and put into practice. Ring-fenced funds and active facilitation by skilled animators would help local actors to implement transformations. In the scope of EUSALP, the Alpine Region Preparatory Action Fund (ARPAF) made available by the European Parliament was extremely helpful to develop a cross-sectorial thematic. This type of funding scheme should be urgently perpetuated, which requires an action by the European Parliament and the Commission. An initiative in this sense is to establish a EUSALP innovation facility, which could mobilize funding from different sources, including public and private funds. At the national level, Switzerland is working with "Modellvorhaben Raumentwicklung", which translates roughly into "models for spatial development". Several federal offices agree on common topics and allocate common funding for them, e.g., for access to public services and digitalization. Another example is the "Ecologic Transition Contract" in France. Other good examples on regional level are the SCORAN and Departmental digital infrastructure schemes and/or digital use schemes in France as well as the Agenda for Sustainable Development (Agenda 2030) of the Metropolitan City of Genova.

R4: Allow room for innovation and experimentation.

Smart Villages is a participatory approach based on local needs identified on the territory. When starting the process, the outcomes are not yet defined. Policies which support the Smart Villages approach must therefore leave room for innovation as an experimentation. They must also allow thematic openness. Good examples are, e.g., the "Zukunftsraum Tirol" in Austria and the strategy for Inner Areas in Italy. The numerous programs for innovation-like Horizon Europe are also very helpful to develop the Smart Villages approach. However, very often, these innovation policies are "territorially blind". They lack a territorial perspective. In addition, programs like Horizon Europe are very difficult to access by "small local players". This should be corrected in order to encourage place-based approaches as Smart Villages.

R5: Encourage networking and the exchange of experiences around the Smart Village approach within rural and mountain areas and with urban areas.

Policies should encourage networking and the awareness raising of relevant stakeholders for digitalization and foster the exchange of know how between stakeholders, e.g., universities and SMEs. The ENRD network on Smart Villages is very helpful at the EU-level. In EUSALP, the creation of a network of Smart Villages is planned for 2021. A good example at the regional level is the project BodenseeMittelstand 4.0 (BoMi 4.0) in Germany, which supports SMEs in their digital transformation by bundling and coordinating with expertise from business, science and SME-related institutions and make it more accessible to regional SMEs. In the same sense, the potential of digitalization must also be used to a greater extent to improve cross-border public services. EUSALP with its multilevel-governance and transnational approach should take up this request and develop appropriate solutions.

R6: Use the potentials of the Smart Villages approach to communicate the innovation potentials and attractiveness of mountain and rural areas and to link urban and rural areas.

With the smart transformation under way or even accomplished, mountain and rural areas can position themselves as being at the forefront of innovation and attractive for residents and new inhabitants. The Smart Village approach helps to develop new business models and job opportunities and with that new economic perspectives for marginalized territories. Good cooperation with the business sector is therefore important. The Smart Village approach also helps to strengthen the resilience of rural villages, as highlighted during the COVID-19 pandemic. These achievements need to be communicated in a clever way, including towards newcomers and young families: Smart Villages offer new opportunities in rural mountainous areas for job creation, innovation and social inclusion and can enhance the quality of life of local communities. Strengthening the linkages between urban and rural areas is also one of the main goals of EUSALP. The EUSALP-perimeter encompasses not only the core alpine area but also the surrounding major urban areas like Lyon, Milan, Ljubljana and Munich. Communication within EUSALP is therefore crucial. On the other hand, cities should also reflect on their connections with the surrounding regions. A good example for communication at regional level is Tirol 2050 (Austria).

R7: Develop digital infrastructures and skills according to the needs and to the technological possibilities.

Good digital infrastructures are a necessary precondition for Smart Villages. Policies that bring forward these infrastructures are urgently needed. Good examples are the National Ultra-Wideband Project in Italy, the Strategic Digital Program of the Liguria Region 2016–2018, the Strategy for Inner Areas in Italy with Its Digital Agenda and the public service obligation in Switzerland, which guarantees a minimum bandwidth of 10 Mbit/s guaranteed for all households and enterprises. Raising this minimum bandwidth to 80 Mbit/s is actually being discussed in the federal parliament. In territories with a failure of market, public investments are necessary to prevent a digital gap. Public investments in these territories must be excluded from the state aid rules. Special attention must be paid to border areas. EUSALP has identified numerous gaps in the fiber optics backbone across borders in the alpine area. These gaps must be filled by a coordinated approach of the national and regional authorities with support from macroregional and EU level. Furthermore, the availability of digital infrastructures tends to be lower in border areas, as infrastructures are often planned and developed from a national or regional perspective. To plan and co-develop digital infrastructures in functional areas across borders should be a task encouraged by EUSALP. The basis could be the foreseen "Common Spatial Development Perspective", which will be elaborated in the years 2021–2022. Digital infrastructures alone are not sufficient. Equally important are the digital skills. The potentials of digitalization can only be used, if the digital skills are well developed. Education, training and coaching are therefore crucial functions which must be strengthened with a clear focus on rural and mountain territories.

## 8. Conclusions

This paper contains an extensive review and synthesis of rural development policies closely related to the concept of Smart Villages. They are divided by six areas identified to be of great importance for rural development: depopulation and population aging, climate change and environmental services, change in production and diversification of the rural economy, infrastructure and basic services, digital transformation and bridging the rural–urban gap and the social aspect of living. For each of these areas existing and future (post 2020) rural policies are presented.

Answering to one of our research goals, we provide an in-depth presentation of these policies. Based on the data collected, we conclude that the most interesting area for improvement is digital transformation and bridging the rural–urban gap. The next area is the area of depopulation and population aging. On the other hand, data related to future policies shows that the area of the social aspect of living is falling behind on the agenda of further development. That is why in the future, the focus should be placed on

the issues of infrastructure and basic services and the social aspect of living, including policies related to education, free time activities, cultural activities and others related to the general well-being of people as they are noticeably lagging behind in comparison to other areas.

The coverage level of the policies can vary between the municipal, regional and national, and from these three, the regional level policy development was the most common. As there is no pattern or scheme that tells us the link between policies and their development levels (municipal, regional, national), as a next step it should be explored which of these levels are proven to work the best for different rural development policies. Furthermore, it should be analyzed which policy characteristics are important in order for policymakers to decide on the coverage level of a specific policy. A common thing amongst the policies regarding their success or the added value was the fact that in policy development, the local need is taken into consideration. In some cases, this means the involvement of the local population in their development, further networking and collaboration during its implementation.

Increased attractiveness for rural villages makes smart transformations in rural areas a necessity for developing sustainable Smart Villages. This is why it is so important that the idea of Smart Villages is integrated as part of the current rural development policies. Another conclusion we can draw is related to the funding problem of Smart Villages. Integrated funding schemes should be established for the development of the Smart Village approach. We also highlight the importance of the development of this approach as a way of communicating the attractiveness of mountain and rural areas, allowing to connect and network with people, as well as opportunities for new employment. To conclude, it is important to note that policies for digital infrastructure are essential conditions for Smart Villages development.

**Author Contributions:** All authors, S.S., P.N., G.L., N.C., A.K., T.E. and E.S.D., have designed the research and the methodology; formal analysis, S.S.; investigation, S.S.; resources, E.S.D., N.C., G.L.; data curation, S.S.; writing—original draft preparation, S.S.; writing—review and editing, S.S., P.N., G.L., N.C., A.K. and E.S.D., supervision, E.S.D.; project administration, E.S.D., P.N.; funding acquisition, E.S.D., A.K., G.L., T.E., P.N. All authors have read and agreed to the published version of the manuscript.

**Funding:** This research was co-funded by the Republic of Slovenia and the European Union from the Regional Development Fund via the Interreg Alpine Space Smart Villages project and the Slovenian Research Agency (core funding ICT4QoL—Information and Communications Technologies for Quality of Life (P2-0246).

**Conflicts of Interest:** The authors declare no conflict of interest.

## Appendix A

**Table A1.** Summary of questionnaire respondents.

| Respondent ID | State | Region | Type of Organization | Type of Partners |
|---|---|---|---|---|
| R1 | Austria | Lower Austria | Public institute | Project partner |
| R2 | Germany | Lake of Constance | Regional authority | Project partner |
| R3 | Switzerland | | NGO | Project partner |
| R4 | Germany | Region Südlicher Obrrhein | Regional authority | Project partner |
| R5 | Slovenia | Podravje | Private institute | Project partner |
| R6 | Austria | Tyrol | Regional authority | Project partner |

**Table A1.** *Cont.*

| Respondent ID | State | Region | Type of Organization | Type of Partners |
|---|---|---|---|---|
| R7 | France | Provence-Alpes-Côte d'Azur/Auvergne Rhône-Alpes | NGO | Project partner |
| R8 | Slovenia | / | National authority | / |
| R9 | Italy | Liguria | ANCI Liguria | Member of Board of Directors of GAL |
| R10 | Switzerland | Region Luzern West | Regional authority | Project partner |
| R11 | Italy | Area Metropolitana di Genova | Local authority | Stakeholder |
| R12 | Italy | Liguria | Local action group | Project partner |
| R13 | Austria | Upper Austria | Research organization | Project partner |

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
