# Peer review of "Smart Villages Policies: Past, Present and Future"

_sustainability, doi:10.3390/su13041663_

Round 1
Reviewer 1 Report
More details about the sample: how many questionnaires have been sent, how many have been collectect and from where, from which type of stakeholders from each region, etc. A table with this information should be introduced, and some analysis about the degree of significance or relevance of collected answers from each region, type of stakeholders or partners, etc.
It seems that some links to references doesn't work (12, 65, 91, etc.). Please check all of them to ensure a proper access. Some other are introduced out of the context of the text or not directly related to the issues you are presenting (ex. 62).
Properly complete references, as number 52: OECD (2018)Rural 3.0. A framework for rural development. On the other hand, is this the same that number 13?.
Introduce references for DESIRA, SHERPA and LIVERUR (line 280)
Reference 62 doesn't work and seems not to be the right one (you are talking on CAP and introduce a reference on Smart Villages). Maybe it should be 63.
Some links are to very generic sites. Reader should find the information just with one click, not to look and navegate through the webs to find the information. Just a couple of examples: The link to the reference number 65 is so generic that it is not useful at all. If you want to show something about the content of rural development in Europe 2020, you should introduce a proper link, not that so generic. And secondly, reference number 116. Same as the last one. There are many others as these ones.
Use the proper references related to the issues raised. For example, What is the relationship with the topic of this paper the reference number 54, related to Vietnan? ¡It is too far away! Please, be sure to use adequate references and not just introduce withouth not much sense.
It doesn't seems adequate an unique section including rural policies literatura as well as survey data collection. The section 5.2 seems to be much more on results (?), and not a review of rural policies literature, as reader can expect by the title. The last one, "survey data collection", should be more related to a section about sources and methodoly, otherways, if the section is related to the analysis of results, use a better title (relating to "results" or similar).
There is some abuse of mentions and short descriptions of EU Research Projects, mixing research and policies (even although those research projects may support policies). This happens in sections 5.2.1 and 5.2.1
In a similar way, in 5.2.2 a lot of information and description is included, and sometimes with insufficient explanation (just as an example, what is it about the National Strategy for Internal Areas in Italy? You told on strategies and actions, planed interventions ... but, what are them about? We don't know).
In a similar way, Age Policy in Region Luzern West: "it deals with demographic development, and improvement of general living conditions for elderly people".
Another example, line 488: Upper Austria: ¡you just said that it is implemented on a regional coverage level!.Lines 574, etc. (curiously, preceed by a good "description" of national Swiss policy development (lines 560-574).
Same for many of the policies announced and shortly and, many times, insuficiently described. Reader need explanations, since paper should be not just an informative list of references.
In order to simplify the massive amount of sections and subsections, authors should consider bigger sections, less informative and more interpretative. In addition, Existing and future policies can be perfectly combined, at least within each of the issues (Depopulation, climate change, etc.).
Discussion section is not a discussion, but just a summary of results.
Policy recommendations are not a -research- discussion. Are just policy recommendations (very important, of course, but not discussion).
Authors said in the conclusions: "this paper contains an extensive review and synthesis of rural development policies closely related to the concept of Smart Villages". This is right, except that through the policy review the Smart village concept is sufficiently present. It seems much more that review and synthesis of policies, but not related to such concept.
Table 1 is not ready. Maybe the summary in the table can allow to significantly reduce the so extensive text describing and giving data.
Reviewer 2 Report
This document contains an extensive review and synthesis of rural development policies linked to the concept of Smart Villages. The link and relations between six large areas identified are very important for rural development, especially for the problems connected to them are evaluated. It is very documentary work but well organized. Very interesting paper.
Reviewer 3 Report
The research is a interesting topic. However, for chapter 5, it will be much more easily to understand if you can make some graphs to support your analysis. For chapter4, for the questionnaire survey, you need to give information about your sample. for example, how many you send and how many feedback.
Round 2
Reviewer 1 Report
Some minor comments have been added through the coverletter, in attachment.

Reviewer 3 Report
It is qualified for publish
Author Response
Dear Reviewer,
Thank you for your approval.
Kind regards,
The authors.